# OpenReview forum: "BYORn: Bootstrap Your Own Responses to Defend Large Vision-Language Models Against Backdoor Attacks"
_ICML.cc/2026/Conference — ICML 2026 regular_

### Official Review · Reviewer_JwmM · 2026-03-10

**Soundness:** 3
**Presentation:** 3
**Significance:** 3
**Originality:** 3
**Overall Recommendation:** 4
**Confidence:** 3

**Summary:**

This paper proposes bootstrap your own responses (BYORn), a defense framework against backdoor attacks during the instruction fine-tuning stage of large vision-language models (LVLMs). The method is built on a key observation: in poisoned datasets, the target responses of backdoor-injected samples exhibit low likelihood (high perplexity) under the pretrained model, as attacker-crafted target responses are often semantically inconsistent with the corresponding image-text input pairs. Extensive experiments are conducted to evaluate the proposed approach across four models, two tasks, and four attack types. Furthermore, the paper designs two adaptive attacks (semantic alignment attack and adversarial perturbation attack) to verify the effectiveness and robustness of BYORn.

**Compliance With Llm Reviewing Policy:**

Affirmed.

**Final Justification:**

Thanks for the authors' rebuttal. After reading it, my concerns were addressed, so I will keep my rating positive.

**Key Questions For Authors:**

1. The paper fixes the threshold at 0.6 (consistently marking 40% of the samples as suspicious). How does BYORn perform when the actual poisoning rate deviates significantly from 10% (e.g., 1% or 50%)? Specifically, under an extremely low poisoning rate, would the erroneous replacement of a large volume of clean samples with model-generated responses lead to a noticeable degradation in performance?

2. Can existing clean-label backdoor attacks (where the target response is semantically plausible but subtly tampered with) bypass the perplexity detection? For instance, an attacker could construct a target response like "The image shows a red car" (when the image does contain a car, but the color has been altered). In this scenario, the perplexity might not increase significantly. Have the authors considered this category of attacks?

**Limitations:**

Yes.

**Strengths And Weaknesses:**

**Strengths**

1. The paper has a clear and well-motivated research direction. As visualized in Figure 3 (histogram), the detection score distributions of clean and poisoned samples exhibit a clear separation, which provides an empirical foundation for the design of the proposed defense method.

2. This work rigorously connects the training objective of BYORn to the minimization of the population risk upper bound. By leveraging the Donsker-Varadhan inequality and Hoeffding's lemma, a formal upper bound is derived, which offers strong theoretical guarantees for the proposed method.

3. The experimental setup is comprehensive, covering 4 distinct model architectures, 4 attack types, 2 downstream tasks, and multiple datasets, alongside comparisons with 2 existing defense methods.

**Weeknesses**

1. Because BYORn needs to "dynamically regenerate responses for suspected poisoned samples (bootstrapping)" during the training process, this introduces a massive computational overhead from autoregressive inference.

2. BYORn's core detector relies entirely on the response likelihood calculated by the pretrained model. Its fundamental premise is that current backdoor attacks often cause the input and target response to be "semantically implausible". If an attacker does not force the model to output completely unrelated words, but instead performs subtle factual tampering or logical induction, the perplexity detection may fail to distinguish the poisoned samples.

3. The method exhibits a certain degree of sensitivity to hyperparameters (the filtering threshold). Table 11 demonstrates the impact of the percentile threshold p. When p=0.6 (the default setting), the CIDEr score is 62.2, and the attack success rate (ASR) is 0. However, if the threshold is slightly lowered to p=0.5, even though the ASR remains at 0, the CIDEr score drops significantly to 54.3

---

> ### Author Rebuttal · Authors · 2026-03-30
>
> We are greatly appreciative for the insightful comments.
> We aim to address the remaining concerns below.
> Also, we kindly ask the reviewer to consider the bugfix we described in response to Reviewer XzKz.
>
> ## W1: Computational overhead
> We acknowledge the computational overhead introduced by autoregressive generation and discuss it in Section 6.3 and Table 7 in the main text.
> We note that this is a common trade-off in backdoor defenses: recent methods for image classification [2,3] and LLMs (Table 5 in [1]) similarly incur overhead.
> Importantly, BYORn introduces no overhead at inference time and can be deployed as any standard fine-tuned model.
>
> Furthermore, the field of efficient generation is rapidly evolving, with promising alternatives to standard autoregressive generation [4], which are likely to further reduce the computational overhead of BYORn.
>
> Nevertheless, in time-critical situations, users can opt for BYORn-F, a lightweight variant of our approach.
> Table 7 update shows that BYORn-F introduces minimal overhead compared to supervised fine-tuning:
>
> |Method|Mem (GB)|Detection (min)|Epoch (min)|ASR (%)|
> |-|-|-|-|-|
> |SFT|180|0|45|99.1|
> |BYORn-F|180|20|27|0.0|
> |BYORn|180|20|135|0.1|
>
> [1] Li, Y., Huang, H., Zhao, Y., Ma, X., & Sun, J. (2025). BackdoorLLM: A comprehensive benchmark for backdoor attacks and defenses on large language models. NeurIPS 2025 Datasets and Benchmarks Track.
> [2] Huang, Kunzhe, et al. "Backdoor defense via decoupling the training process." arXiv:2202.03423 (2022).
> [3] Gao, Kuofeng, et al. "Backdoor defense via adaptively splitting poisoned dataset." CVPR 2023.
> [4] Khanna, S., et al. (2025). Mercury: Ultra-fast language models based on diffusion. arXiv-2506.
>
> ## W2: Semantically plausible attacks
> We acknowledge that more sophisticated attacks may emerge that could reduce the effectiveness of BYORn's detection step.
> In parallel, we expect the development of new defenses and detection methods, which will be easy to integrate into BYORn due to its modular design.
> Concretely, the detection step could be replaced with a stronger, more robust detector as such methods become available.
> This ensures BYORn remains effective as the field evolves.
> We briefly discuss this in the limitations section and will elaborate further in the next revision.
>
> Nevertheless, we implemented a more semantically plausible attack and evaluated BYORn against it.
> See our response to Q2 for details.
>
> ## W3: Filtering threshold
> We measure sensitivity to the percentile threshold `p` by extending Table 11 with additional experiments.
> The results indicate that BYORn consistently leads to a robust model that outputs responses that are semantically aligned with the ground truth.
> Both SPICE and CIDEr drop between `p=0.5` and `p=0.6`.
> We hypothesize that most useful training samples are within this 10% which BYORn relabels, leading to lower accuracy.
>
> ||p=0.3|p=0.5|p=0.6|p=0.9|p=0.95|p=0.975|
> |-|-|-|-|-|-|-|
> |ASR↓|0|0|0|0|1.1|1.2|
> |CIDEr↑|55.6|55.2|62.2|61.4|60.4|59.8|
> |SPICE↑|19.5|16.7|18|18.2|18.9|20.1|
> |Recall|1|1|1|0.998|0.5|0.25|
> |#FN|0|0|0|3|1162|1743|
> |#FP|13944|9296|6972|3|0|0|
>
> Please refer to tables we attached to Reviewer fxJJ for more insights.
>
> ## Q1: High and low poisoning rates
> We evaluate BYORn (LlaVA-1.5 under VL-Trojan) at poisoning rates of 1%, 10%, and 50%.
>
> |Poisoning rate|CIDEr|SPICE|ASR (%)|
> |-|-|-|-|
> |1%|62.0|18.4|0.0|
> |10%|62.5|18.2|0.0|
> |50%|61.5|18.2|0.2|
>
> We observe consistent CIDEr and SPICE across all poisoning rates.
> BYORn remains robust even at 50% poisoning rate, where approximately 10% of poisoned samples survive filtering and are included in training with their original, poisoned responses.
> This is because BYORn replaces the poisoned responses of most of the poisoned examples, thereby breaking the trigger-target correlation.
>
> ## Q2: Clean-label attacks
> During rebuttal period, we designed a clean-label attack similar to the example suggested by the Reviewer.
> Concretely, we selected all training examples containing "car" in their caption (~7% of the dataset), applied the Blend trigger to those images, and replaced their captions with the target response: "The image shows a red car."
> While not a fully clean-label attack, the target response is far more semantically plausible than in standard dirty-label settings.
> We measure attack success as the fraction of generated responses containing "red car."
> We present our preliminary results in the table below:
>
> ||SFT|BYORn-F|BYORn|
> |-|-|-|-|
> |CIDEr/ASR|59.0/55.2|40.2/0.0|60.0/0.1|
>
> Both BYORn and BYORn-F achieve complete robustness against this attack.
> Even a semantically plausible target response retains enough misalignment with the image to be detected by our likelihood-based score.
> Pushing the target response closer to the true image content improves stealthiness but weakens the attack signal, limiting the attack threat.
>
> We leave the design of a fully clean-label attack for VLMs as an open challenge for future work.

---

> > ### Author Rebuttal · Reviewer_JwmM · 2026-04-02
> >
> > Thanks for the authors' rebuttal. After reading it, my concerns were addressed, so I will keep my rating positive.

---

> > > ### Author Response · Authors · 2026-04-07
> > >
> > > Thank you for taking the time to read our rebuttal and for your thoughtful comments throughout the review process. Your feedback has been very helpful in improving the paper.

---

### Official Review · Reviewer_JNS9 · 2026-03-11

**Soundness:** 3
**Presentation:** 3
**Significance:** 3
**Originality:** 2
**Overall Recommendation:** 4
**Confidence:** 4

**Summary:**

This paper proposes BYORn, a backdoor-robust instruction-tuning framework for large vision-language models that detects suspicious, low-likelihood target responses and replaces them with EMA-generated responses during fine-tuning. Extensive experiments show that BYORn improves robustness to backdoor attacks while preserving clean performance.

**Compliance With Llm Reviewing Policy:**

Affirmed.

**Key Questions For Authors:**

What fraction of benign samples are actually bootstrapped or discarded?  What fraction of malicious samples are undetected?

**Limitations:**

yes

**Strengths And Weaknesses:**

**Strengths**
1. The paper targets the problem of backdoor robustness for open-ended LVLM fine-tuning and proposes a simple defense that is easy to understand.
2. The empirical section is fairly broad within the chosen scope, covering multiple attacks and several LVLM families.

**Weaknesses**
1. **The paper does not follow standard ICML formatting (e.g., table captions are placed below the tables rather than above them).**
2. Can BYORn be extended to broader instruction-following settings, such as VQA? For example, BYE (Rong et al., 2025) tested performance against backdoor attacks on ScienceQA, IconQA, and RSVQA.
3. Lack of failure case analysis. What kind of backdoor attack would render the defense ineffective? Can "label-consistent" or "highly plausible" poisoned targets break through your defense?

---

> ### Author Rebuttal · Authors · 2026-03-30
>
> We are greatly appreciative for the insightful comments.
> We aim to address the remaining concerns below.
> Also, we kindly ask the reviewer to consider the bugfix we described in response to Reviewer XzKz.
>
>
> ## W1: Table captions below rather than above
>
> We apologize for the oversight and thank the reviewer for pointing this out. We will move all table captions above the tables in the next revision.
>
> ## W2: VQA experiments
>
> As suggested by the reviewer, we conducted VQA experiments on the ScienceQA benchmark following the setup from BYE (Rong et al., 2025). Our approach achieves complete backdoor resilience against four different attacks. For more details, please refer to our response on Weakness 2 of the Reviewer fxJJ.
>
> ## W3: Lack of failure case analysis
>
> Label-consistent attacks have been successfully deployed against classification models [1].
> However, implementing such attacks in the instruction fine-tuning setting is non-trivial due to the open-ended target space, and to the best of our knowledge, remains unexplored.
> We expect such attacks to be more challenging for our method, as it relies on the input-response semantic gap to detect poisoned examples.
> Related to this discussion, we implemented a "highly plausible" attack suggested by another reviewer and evaluated BYORn against it. For more details, please refer to our response to Question 2 from Reviewer JwmM.
>
> We thank the reviewer for this remark. We will include a more detailed analysis of the potential failure cases in the next version of the manuscript, along with an extended discussion of the method limitations.
>
> [1] Turner, A., Tsipras, D., & Madry, A. (2019). Label-consistent backdoor attacks. arXiv preprint arXiv:1912.02771.
>
> ## Q1: Detection metrics
> Hyperparameter `p` dictates how much samples BYORn-F and BYORN discard or bootstrap, respectively.
> We mark undetected malicious samples as false negatives and measure recall on extended Table 11:
>
> | | p=0.3 | p=0.5 | p=0.6 | p=0.9 | p=0.95 | p=0.975 |
> |-|-|-|-|-|-|-|
> | ASR↓ | 0 | 0 | 0 | 0 | 1.1 | 1.2 |
> | CIDEr↑ | 55.6 | 55.2 | 62.2 | 61.4 | 60.4 | 59.8 |
> | SPICE↑ | 19.5 | 16.7 | 18 | 18.2 | 18.9 | 20.1 |
> | Recall | 1 | 1 | 1 | 0.998 | 0.5 | 0.25 |
> | #FN | 0 | 0 | 0 | 3 | 1162 | 1743 |
> | #FP | 13944 | 9296 | 6972 | 3 | 0 | 0 |
>
> Please refer to tables we attached to Reviewer fxJJ for more insights.

---

> > ### Author Rebuttal · Reviewer_JNS9 · 2026-04-02
> >
> > The above rebuttal has addressed my concerns.

---

> > > ### Author Response · Authors · 2026-04-07
> > >
> > > Thank you for taking the time to read our rebuttal and for your thoughtful comments throughout the review process. Your feedback has been very helpful in improving the paper.

---

### Official Review · Reviewer_fxJJ · 2026-03-13

**Soundness:** 3
**Presentation:** 3
**Significance:** 3
**Originality:** 3
**Overall Recommendation:** 4
**Confidence:** 3

**Summary:**

This paper studies backdoor attacks in supervised fine-tuning for large vision-language models. The main idea is that poisoned target responses are often a poor semantic match for the image and instruction, so a pretrained model gives them low likelihood. Based on this, the paper proposes two methods: BYORn-F, which filters suspicious training examples, and BYORn, which replaces suspicious target responses with responses generated by the model during training. The paper also gives a theoretical motivation for this training objective through an empirical upper bound on clean-data risk. Experiments on image captioning and spot-the-difference, using several LVLMs and multiple attack types, show that BYORn usually achieves much lower attack success rate than prior defenses while keeping competitive clean-task performance, including under adaptive attacks.

**Compliance With Llm Reviewing Policy:**

Affirmed.

**Final Justification:**

The rebuttal has addressed my main concerns, and I maintain my prior assessment.

**Key Questions For Authors:**

1. Can you provide a more detailed analysis of detector false positives and false negatives across attack types, and explain how much BYORn depends on high recall versus high precision?

2. Have you tested, or do you have evidence that the method transfers to broader multimodal instruction-tuning tasks beyond captioning and spot-the-difference, such as VQA, chart understanding, or multimodal dialogue?

**Limitations:**

yes.

**Strengths And Weaknesses:**

Strengths

1. The problem is important and interesting. Backdoor robustness for open-ended multimodal generation is still under-studied, and the paper explains clearly why many existing defenses do not transfer well from classification to LVLM instruction tuning.

2. The  idea is simple and intuitive. Using response likelihood to detect poisoned samples is easy to understand, and the full BYORn method improves on plain filtering by keeping suspicious samples but rewriting their targets, which is a meaningful step beyond sample removal. The method is also presented as model-agnostic and trigger-agnostic, which increases its practical value.

Weaknesses

1. The method depends heavily on the assumption that poisoned responses are unlikely under the pretrained model. This is reasonable for current attacks, but it may become weaker when the attacker makes the poisoned response more semantically plausible.

2. It would be stronger to also test more general instruction-following settings such as VQA or multimodal chat.

---

> ### Author Rebuttal · Authors · 2026-03-30
>
> We are greatly appreciative for the insightful comments.
> We aim to address the remaining concerns below.
> Also, we kindly ask the reviewer to consider the bugfix we described in response to Reviewer XzKz.
>
> ## W1: Poisoned responses assumption
>
> We acknowledge that more sophisticated attacks may emerge that could reduce the effectiveness of our detection approach. In parallel, we expect the development of new defenses and detection methods, which will be easy to integrate into BYORn due to its modular design. This allows BYORn to remain relevant as the field evolves.
>
> Nevertheless, prompted by reviewer JwmM, we have implemented one concrete semantically plausible attack and evaluated BYORn against it. For more details, please refer to our response to Question 2 from reviewer JwmM.
>
> ## W2 & Q2: VQA experiments
>
> We conducted VQA experiments on the ScienceQA benchmark following the setup from BYE (Rong et al., 2025). The tables below report accuracy and attack success rate (ACC/ASR), averaged over four attacks (BadNets, Blend, DualKey, VL-Trojan), for Qwen3-VL and LLaVA-1.5 models across five different defenses.
> We observe that both BYORn-F and BYORn achieve complete backdoor resilience across all attacks on ScienceQA, with ASR dropping to 0% in all settings.
> With Qwen3-VL, this robustness comes at virtually no cost to accuracy, matching the clean performance baseline.
> With LLaVA-1.5, both methods remain fully robust, though we observe a moderate drop in accuracy compared to the undefended model.
>
> | Attack     | No Defense | ONION | BYE | BYORn-F | BYORn |
> |------------|------------|-------|-----|---------|-------|
> | **Qwen3-VL**   |            |       |     |         |       |
> | BadNets    | 89.7 / 61.3 | 89.7 / 61.3 | 90.5 / 0.0 | 88.9 / 0.0 | 86.7 / 0.0 |
> | Blend      | 89.6 / 72.7 | 89.6 / 72.7 | 90.9 / 75.9 | 89.2 / 0.0 | 86.7 / 0.0 |
> | DualKey    | 89.3 / 92.0 | 89.3 / 0.1 | 89.5 / 74.5 | 88.4 / 0.0 | 87.2 / 0.0 |
> | VL-Trojan  | 89.9 / 92.4 | 89.9 /0.1 | 90.3 / 0.0 | 89.1 / 0.0 | 86.8 / 0.0 |
> | **Average** | **89.6 / 79.6** | **89.6 / 33.6** | **90.3 / 37.6** | **88.9 / 0.0** | **86.9 / 0.0** |
>
> | Attack     | No Defense | ONION | BYE | BYORn-F | BYORn |
> |------------|------------|-------|-----|---------|-------|
> | **LLaVA-1.5**  |            |       |     |         |       |
> | BadNets    | 80.2 / 100.0 | 80.2 / 100.0 | 77.2 / 98.6 | 68.3 / 0.0 | 64.6 / 0.0 |
> | Blend      | 80.6 / 100.0 | 80.6 / 100.0 | 76.8 / 100.0 | 68.4 / 0.0 | 65.5 / 0.0 |
> | DualKey    | 79.9 / 100.0 | 79.9 / 0.0 | 76.5 / 100.0 | 67.8 / 0.0 | 66.5 / 0.0 |
> | VL-Trojan  | 80.4 / 100.0 | 80.4 / 0.0 | 74.7 / 100.0 | 67.4 / 0.0 | 65.4 / 0.0 |
> | **Average** | **80.3 / 100.0** | **80.3 / 50.0** | **76.3 / 99.7** | **68.0 / 0.0** | **65.5 / 0.0** |
>
> ## Q1: Detector metrics
> We extend Table 11 and validate that BYORn remains robust when recall falls under 100%.
> We count false positives (FP) as clean samples marked suspicious, and undetected poisoned samples as false negatives (FN).
> We validate LLaVA under all four attacks when training and evaluating Flickr30:
>
> |**⚠Blend**|p|CIDEr↑|ASR↓|SPICE↑|P|R|#FN|#FP|
> |-|-|-|-|-|-|-|-|-|
> |BYORN|0.3|55.6|0|19.5|0.14|1|0|13944|
> |BYORN|0.5|55.2|0|16.7|0.2|1|0|9296|
> |BYORN|0.6|62.2|0|18|0.25|1|0|6972|
> |BYORN|0.9|61.4|0|18.2|0.998|0.998|3|3|
> |BYORN-F|0.95|52.4|57.6|18.9|1|0.5|1162|0|
> |BYORN|0.95|60.4|1.1|18.9|1|0.5|1162|0|
> |BYORN-F|0.975|53.7|79.3|20.1|1|0.25|1743|0|
> |BYORN|0.975|59.8|1.2|20.1|1|0.25|1743|0|
>
> |**⚠DualKey**|p|CIDEr↑|ASR↓|SPICE↑|P|R|#FN|#FP|
> |-|-|-|-|-|-|-|-|-|
> |BYORN|0.6|62.1|0|18.3|0.25|1|0|6972|
> |BYORN|0.9|62.5|0|19|0.999|0.999|1|1|
> |BYORN-F|0.95|54.9|0.5|19.8|1|0.5|1162|0|
> |BYORN|0.95|62.5|0.1|18.9|1|0.5|1162|0|
> |BYORN-F|0.975|56.7|30.7|19.9|1|0.25|1743|0|
> |BYORN|0.975|62|0.3|19.9|1|0.25|1743|0|
>
> |**⚠VL-Trojan**|p|CIDEr↑|ASR↓|SPICE↑|P|R|#FN|#FP|
> |-|-|-|-|-|-|-|-|-|
> |BYORN|0.6|62.9|0|18.3|0.25|1|0|6972|
> |BYORN|0.9|61.6|0.1|19.3|0.99|0.99|3|3|
> |BYORN-F|0.95|54.2|11|20.2|1|0.5|1162|0|
> |BYORN|0.95|62.2|0.4|18.6|1|0.5|1162|0|
> |BYORN-F|0.975|54.6|64.8|19.9|1|0.25|1743|0|
> |BYORN|0.975|60.6|2.4|19.3|1|0.25|1743|0|
>
> The extended results demonstrate that our method is not sensitive to the percentile threshold `p`.
> Furthermore, at highest `p` the full BYORn outperforms BYORn-F due to the ability to relabel suspicious poisoned samples.

---

> > ### Author Rebuttal · Reviewer_fxJJ · 2026-04-02
> >
> > Thanks for the authors’ efforts. My concerns have been addressed.

---

> > > ### Author Response · Authors · 2026-04-07
> > >
> > > Thank you for taking the time to read our rebuttal and for your thoughtful comments throughout the review process. Your feedback has been very helpful in improving the paper.

---

### Official Review · Reviewer_XzKz · 2026-03-15

**Soundness:** 3
**Presentation:** 3
**Significance:** 3
**Originality:** 3
**Overall Recommendation:** 4
**Confidence:** 3

**Summary:**

This paper proposes BYORn, a backdoor-robust fine-tuning framework for large vision-language models (LVLMs). It addresses the vulnerability of supervised fine-tuning to backdoor attacks, where an adversary poisons a fraction of the training dataset by injecting triggers and corrupting corresponding labels. The paper is well-organized, and experiments demonstrate strong performance over baseline methods.

**Compliance With Llm Reviewing Policy:**

Affirmed.

**Final Justification:**

Thanks for the authors' rebuttal. My concerns have been addressed, so I will keep my rating positive.

**Key Questions For Authors:**

1. The proposed method appears to rely heavily on the quality of the pretrained model to distinguish clean samples from poisoned ones. If the pretrained model is itself unable to reliably detect or differentiate backdoored data, it is unclear how the framework would behave.

2. It would be helpful if the authors could provide a more detailed description of how the adaptive attack is implemented — specifically, how the attack is designed to minimize the objective in Eq. 3. A clearer explanation of the adaptive attack setup would make it easier to assess the robustness ot proposed method.

**Limitations:**

See question above

**Strengths And Weaknesses:**

1. The paper tackles an important and practical problem: fine-tuning VLMs in the presence of poisoned training data. The proposed backdoor-resilient training framework, which learns from potentially poisoned data is well-motivated and technically interesting.

2. The paper provides theoretical guarantees to support the proposed method, which strengthens the credibility of the framework beyond empirical evaluation alone.

3. The paper is clearly structured and easy to follow.

---

> ### Author Rebuttal · Authors · 2026-03-30
>
> We are greatly appreciative for the insightful comments. We aim to address the remaining concerns below. Also, we kindly ask the reviewer to consider the bugfix we described below.
>
> ## Note on bugfix
>
> Since our initial submission, we noticed an implementation discrepancy from Eq (3). The code computed log-perplexity over inputs and the target response instead of responses only.
> This affected results with:
> - negligible change in accuracy and
> - a noticeable ASR improvement in Qwen3-VL against DualKey experiment.
>
> New Table 1 results for Qwen3-VL:
> ||Flickr ASR↓(Δ)|COCO ASR↓(Δ)|
> |-|-|-|
> |BYORn-F|0.9 (−15.9)|2.3 (−10.8)|
> |BYORn|2.1 (−7.8)|2.7 (−3.9)|
>
> In summary, the fix exclusively affects the Qwen3-VL DualKey setting, where it leads to a noticeable improvement, while all other results show only negligible changes.
>
> Our next draft is ready with the new results.
> ## Q1: Detection of poisoned examples
>
> Detection of poisoned examples is an integral stage of our framework. Our approach assumes higher log-perplexity for poisoned examples, which we find reasonable for many domains and attacks due to the large-scale pretraining of the vision-language model. However, we acknowledge that our current detection approach might be less effectve in some  domains. Importantly, BYORn is a modular framework, allowing the detection stage to be enhanced as more robust methods become available. We briefly discuss this in the limitations section and will elaborate further upon acceptance.
>
> ## Q2: Details of the adaptive attack
>
> The goal of this attack is to design a trigger that increases the likelihood of the poisoned response under a pretrained model.
> Since our detector flags samples with low likelihood (Eq. 3), increasing the likelihood causes the poisoned sample to blend in with clean examples and evade filtering.
> To do so, the attacker uses gradient descent to find a small image perturbation that makes the model agree with the poisoned response.
> Therefore, the poisoned image looks like a natural input for which the poisoned response would be the correct answer.
>
> Concretely, for each poisoned sample $(\mathbf{x}_i, \mathbf{q}_i, \mathbf{y}_i)$, the attacker uses PGD  [1] to find an $\epsilon$-bounded perturbation $\boldsymbol{\delta}_i$ that minimizes the loss of the poisoned response under the pretrained model:
>
> $\boldsymbol{\delta}\_{i} =  \arg\min_{\|\boldsymbol{\delta}\|_\infty \leq \epsilon} \mathcal{L}(\theta \mid \mathbf{x}_i + \boldsymbol{\delta}, \mathbf{q}_i, \mathbf{y}_i)$
>
> where $\mathcal{L}(\theta \mid \mathbf{x}_i + \boldsymbol{\delta}, \mathbf{q}_i, \mathbf{y}_i)$ is the detection score (i.e. autoregressive negative log likelihood) defined in Eq. 3 of the main text.
>
> We will update L354-L358 (right column) in the main text and the text in Appendix D to make this clearer.
>
> [1] Madry, Aleksander, et al. "Towards deep learning models resistant to adversarial attacks." arXiv preprint arXiv:1706.06083 (2017).

---

> > ### Author Rebuttal · Reviewer_XzKz · 2026-04-02
> >
> > Thank you for the response. My concerns have been addressed.

---

> > > ### Author Response · Authors · 2026-04-07
> > >
> > > Thank you for taking the time to read our rebuttal and for your thoughtful comments throughout the review process. Your feedback has been very helpful in improving the paper.

---

### Decision · Program_Chairs · 2026-04-30

**Decision:**

Accept (regular)

**Comment:**

This paper proposes a backdoor-robust fine-tuning framework for vision-language models that detects low-likelihood poisoned responses and replaces them during training . Reviewers find the approach well-motivated, with solid empirical results and supporting theory. Concerns mainly relate to reliance on pre-trained likelihood for detection, computational cost, and robustness to more advanced attacks. The authors provided thorough rebuttal with additional experiments and clarifications, which largely addressed these concerns, and reviewers maintained positive scores.

After reading the paper and rebuttal, the AC recommends acceptance of the paper.